# Non-Coding RNAs in Preeclampsia—Molecular Mechanisms and Diagnostic Potential

**DOI:** 10.3390/ijms221910652

**Published:** 2021-09-30

**Authors:** Jelena Munjas, Miron Sopić, Aleksandra Stefanović, Rok Košir, Ana Ninić, Ivana Joksić, Tamara Antonić, Vesna Spasojević-Kalimanovska, Uršula Prosenc Zmrzljak

**Affiliations:** 1Department of Medical Biochemistry, Faculty of Pharmacy, University of Belgrade, Street Vojvode Stepe 450, 11000 Belgrade, Serbia; jelenaj@pharmacy.bg.ac.rs (J.M.); miron@pharmacy.bg.ac.rs (M.S.); aleksandra.stefanovic@pharmacy.bg.ac.rs (A.S.); ana.ninic@pharmacy.bg.ac.rs (A.N.); tamara.antonic@pharmacy.bg.ac.rs (T.A.); vkalima@pharmacy.bg.ac.rs (V.S.-K.); 2BIA Separations CRO, Labena Ltd., Street Verovškova 64, 1000 Ljubljana, Slovenia; rok.kosir@labena.si; 3Genetic Laboratory Department, Obstetrics and Gynaecology Clinic “Narodni Front”, Street Kraljice Natalije 62, 11000 Belgrade, Serbia; ivanajoksic@yahoo.com

**Keywords:** preeclampsia, microRNA, lncRNA, biomarkers

## Abstract

Preeclampsia (PE) is a leading cause of maternal and neonatal morbidity and mortality worldwide. Defects in trophoblast invasion, differentiation of extravillous trophoblasts and spiral artery remodeling are key factors in PE development. Currently there are no predictive biomarkers clinically available for PE. Recent technological advancements empowered transcriptome exploration and led to the discovery of numerous non-coding RNA species of which microRNAs (miRNAs) and long non-coding RNAs (lncRNAs) are the most investigated. They are implicated in the regulation of numerous cellular functions, and as such are being extensively explored as potential biomarkers for various diseases. Altered expression of numerous lncRNAs and miRNAs in placenta has been related to pathophysiological processes that occur in preeclampsia. In the following text we offer summary of the latest knowledge of the molecular mechanism by which lnRNAs and miRNAs (focusing on the chromosome 19 miRNA cluster (C19MC)) contribute to pathophysiology of PE development and their potential utility as biomarkers of PE, with special focus on sample selection and techniques for the quantification of lncRNAs and miRNAs in maternal circulation.

## 1. Introduction

Hypertensive pregnancy complications, including preeclampsia (PE), are one of the most common direct causes of maternal and fetal morbidity and mortality [1]. PE is defined as a new-onset hypertension, diagnosed after the 20th week of gestation, which solves postpartum, followed by proteinuria or some other form of end-organ damage: thrombocytopenia, impaired liver function, development of renal insufficiency, pulmonary edema, or cerebral and visual disturbances [2]. PE is a two-stage syndrome [3]. The first stage begins early in pregnancy, and is associated with impaired trophoblast differentiation, invasion and ultimately remodeling of the spiral arteries, resulting in an inadequate transition of the blood vessel phenotype from small-diameter and high-resistance blood vessels into blood vessels of high capacity and low resistance [3,4,5]. In healthy pregnancy, remodeling of the spiral arteries ensures adequate blood flow to the uteroplacental unit [4,5]. In PE, trophoblast invasion is limited only to spiral arteries in the superficial decidua, resulting in hemodynamic disturbances, leading to poor uterine-placental blood flow and placental hypoxia [4,6]. It should be emphasized that difference in the etiopathogenesis of early-onset (EO, developed before the 34th week of gestation) and late-onset (LO, developed after the 34th week of gestation) PE exists [7]. EO-PE is referred to as a fetal disorder associated with placental dysfunction (previously described as incomplete trophoblast invasion). On the other side, the LO syndrome is thought to be a disorder predisposed by specific maternal risk factors, such as obesity, diabetes, or chronic hypertension [7,8,9], where limited blood flow between the chorionic villi during placental maturation may lead to placental hypoxia [9].

In both cases, resulting hypoxia and ischemia/reperfusion lead to an increased production of humoral factors from trophoblast cells together with the sequestration of the particles from the surface of the injured syncytium of the human placenta, and their subsequent release into the uteroplacental circulation [10]. In healthy pregnancy, trophoblast derived extracellular vesicles (EV) are packed with specific cargo of humoral factors and miRNAs, which can influence endothelial cell function and maternal immune cells (monocytes, granulocytes, T cells and natural killer (NK) cells), having in that way a major effect on immune tolerance to the developing fetus and on the development and function of the placenta [11,12,13,14,15]. In PE, the composition of the cargo carried in the vesicles is altered, with the number of the vesicles being increased, which significantly contributes to the pathophysiology of the disease [16,17,18,19,20,21,22]. Maternal endothelium may be vulnerable to the products of trophoblast cells, which further stimulate cytokine production, adhesion molecules and neutrophils migration into blood vessels walls and generate systemic inflammatory response which results with clinical manifestations of the disease (the second stage of PE) [23,24]

The biggest challenge regarding PE is clear understanding of the disease etiopathogenesis, which entails limited therapeutic approach and lack of effective clinical and biochemical multi-marker algorithms for preeclampsia prediction [4].

Recent technological advancements empowered transcriptome exploration and led to the discovery of numerous non-coding RNA species (small nuclear RNA, small nucleolar RNAs, telomerase RNA, tRNA-derived fragments, tRNA halves, microRNA, small interfering RNA, piwi-interacting RNA, enhancer RNA, long non-coding RNA, and circular RNA), of which microRNAs (miRNAs) and long non-coding RNAs (lncRNAs) are the most investigated [25].

A growing body of evidence supports the involvement of placenta expressed miRNAs and lncRNAs in the regulation of crucial processes of placenta development and function. The complete biological roles of the most miRNAs and lncRNA in placenta remain scarce, nevertheless, it has been unequivocally shown that altered expression of numerous miRNAs as well as lncRNAs in placenta, is related to pathophysiological processes that occur in PE [25,26,27,28,29,30].

The aim of this review is to summarize the latest knowledge of the molecular mechanisms by which miRNAs (focusing on the chromosome 19 miRNA cluster (C19MC)) and lnRNAs contribute to pathophysiology of PE and to review the latest evidence of their potential utility as biomarkers of PE.

## 2. MicroRNAs

MiRNAs are small, about 22–24 nucleotides long, noncoding RNAs that regulate gene expression at post transcriptional level. Half of all currently identified miRNAs are transcribed mostly from introns and few exons of protein coding genes, while the remaining are intergenic, transcribed independently of a host gene [31,32]. Their biogenesis starts with the processing of RNA polymerase II/III transcripts post- or co-transcriptionally [33]. In the canonical, dominant pathway of miRNA biogenesis, primary miRNAs (pri-miRNAs) are transcribed from their genes and processed into double-stranded precursors (pre-miRNAs), by the microprocessor complex: RNA binding protein DiGeorge Syndrome Critical Region 8 (DGCR8) and a ribonuclease III enzyme, Drosha [34]. DGCR8 recognizes an N6-methyladenylated GGAC and other motifs within the pri-miRNA [35], while Drosha cleaves the pri-miRNA duplex at the base of the characteristic hairpin structure of pri-miRNA. Pre-miRNAs are exported to the cytoplasm by an exportin 5 (XPO5)/RanGTP complex and further processed by the RNase III endonuclease Dicer [34,36], which results in the removal of the terminal loop, yielding a mature miRNA duplex [33,37]. This step is followed by unwinding of mature miRNA duplex and loading into the Argonaute2 (Ago2)-containing, RNA-induced silencing complex (RISC) [38,39], where the miRNAs repress mRNAs in a sequence-specific manner [40]. The directionality of the miRNA strand determines the name of the mature miRNA form (5p of 3p derived from the 5′ end and the 3′ end, respectively). Both strands derived from the mature miRNA duplex can be loaded into RISC [38,39], depending on their thermodynamic stability at the 5′ ends of the miRNA duplex or a 5′ U at nucleotide position 1 [39]. The loaded strand is called the guide strand, while the unloaded strand is called the passenger strand, which will be unwound from the guide strand through various mechanisms and cleaved by AGO2 and degraded by cellular machinery [33]. The miRNA biogenesis pathway is shown in Figure 1. In addition, multiple non-canonical miRNA biogenesis pathways exist, using different combinations of the proteins involved in the canonical pathway, mainly Drosha, Dicer, exportin 5, and AGO2 [39].

In most cases, miRNAs interact with the 3′ UTR of target RNAs to suppress expression [33,39], promoting their degradation or inhibiting translation; in others, interaction with 5′ UTR and coding part of target sequence and gene promoters occurs, while under certain conditions, miRNAs can even activate gene expression [39]. Recent studies have suggested that subcellular compartmentation of miRNAs has an effect on the rate of translation, and even transcription [41]. By these mechanisms, miRNAs regulate more than 30–60% of protein coding genes in the human genome [39,42]. MiRNAs are now recognized as pivotal posttranscriptional regulators in a broad range of biological processes [43]. During embryogenesis, critical periods are controlled by epigenetic modification of genes: gamete development, preimplantation embryo development, as well as placentation [44].

### 2.1. Placental MiRNAs

Key molecules involved in miRNA biogenesis, such as Drosha, Exportin 5, Dicer, Ago2 and DEAD-box helicase protein (DP103), have been identified in placental trophoblast cells [45,46]. In Ago2 mutant mouse miRNA machinery is disabled which causes malformation of placental labyrinthine layer with a greatly reduced thickness and death of the mutant embryos at midgestation, highlighting the essential role that miRNAs have in placental development and function [47].

So far, more than 600 miRNAs have been identified in human placenta, including miRNAs originating from imprinted gene clusters, many of which are predominantly or even solely expressed in placenta [48,49]. The imprinted genes are usually activated at critical developmental stages and are involved in cell differentiation, embryonic and placental growth and regulation of nutritional requirements [15,49]. The chromosome 19 microRNA cluster (C19MC) is located at the imprinted, paternally inherited allele on the human chromosome Chr19q13 site and represents one of the largest human miRNA clusters. It is composed of 46 pre-miRNA genes, yielding 59 mature miRNAs [49,50]. C19MC is expressed almost exclusively in placenta, with miRNAs originating from C19MC being the most abundant miRNAs expressed in human term trophoblasts [51]. High C19MC expression is now even considered as one of the most robust markers to define primary first-trimester trophoblast cells [52], while in situ hybridization revealed that the main source of these miRNAs is the trophoblast layer with dominant signals in the syncytiotrophoblast [53,54]. The expression of certain miRNAs belonging to C19MC, at far lower levels than placenta, is also detected in the germ cells, embryonic stem cells (ESC) and certain tumors [39,40,55]. The C19MC is found only in primates and is highly enriched in Alu elements [50]. The expression of C19MC is controlled by methylation at the upstream CpG rich promoter region [50]. Fothi et al. described an epigenetically regulated placental tissue-specific promoter, which is silenced in stem cells. This promoter is particularly efficient in attracting the transcription protein complex that optimizes cluster expression [56]. C19MC miRNAs are transcribed by RNA polymerase II and are processed from introns of a poorly characterized transcript, C19MC-HG (host gene), composed of many repeated non-coding exons [57].

Chromosome 14 microRNA cluster (C14MC), located at the imprinted, maternally inherited allele at chromosome 14q32 site, is composed of 52 miRNA genes, most of them originating from two genomic regions: the miR-127/miR-136 cluster and the miR-379-miR-410 cluster [39,40,49]. C14MC is regulated by methylation of a distal Intergenic Germline derived Differentially Methylated Region [58]. Members of C14MC are not placenta specific but are predominantly expressed in placenta and epithelial tissues [59], with high levels of expression in the adult brain [49].

The miR-371-3 cluster is located on chromosome 19 within a 1050 bp region approximately 20 kb downstream of the C19MC [60]. It is predominantly expressed in the placenta [61] however; some members of the miR-371-3 cluster are highly expressed in human ESCs [62]. The miR-371-3 cluster in humans is composed of hsa-miR-371a-3p, hsa-miR-372 and hsa-miR-373-3p, which all share the same seed sequence “AAG UGC”. Along with them, hsa-miR-371-5p and hsa-miR-373-5p, synthesized from the opposite site of the pre-microRNA, and hsa-miR-371b-3p, are also a part of the miR-371-3 cluster [40]. The miR-371-3 cluster appears to be important for maintaining cell cycle progression, regulation of proliferation, and controlling apoptosis [63].

The expression of the C19MC cluster is detected as early as 5 weeks of pregnancy, and markedly increases in placental trophoblasts from the first to the third trimester [40,64,65]. In contrast, the expression levels of the C14MC are the highest in the first trimester placenta, and decrease during the course of pregnancy [40,64,65]. The expression of the 371–373 cluster only slightly changes during pregnancy [40,65].

Along with these three clusters, many other miRNAs are found to be expressed in placenta and influence the development and function of the organ, however, they are abundantly expressed in many other human cells and therefore are not placenta specific [48,49]. In further text, we will focus on the roles of the largest, placenta specific C19MC in PE.

### 2.2. C19MC MiRNAs and Trophoblast Differentiation, Invasion and Angiogenesis

Several in vitro studies indicated that C19CM is implicated in crucial processes required for adequate placentation, and that their dysregulation has an impact on trophoblast differentiation, invasion and angiogenesis. It was shown that C19MC miRNAs selectively attenuate migration of human trophoblasts by regulating target transcripts related to cellular movement. In particular, miR-519d regulated the extravillous trophoblast (EVT) invasive phenotype by targeting CXCL6, NR4A2 and FOXL2 transcripts through a 3′UTR miRNA-responsive element [66]. MiR-519d-3p directly targeted the 3‘UTR of matrix metalloproteinase-2 (MMP-2), consequently suppressing trophoblast invasion and migration [67]. Another member of the C19MC, miR-520g also targeted MMP2, and suppressed migration and invasion of HTR-8/SVneo cells [68]. Mir515-5p was upregulated in PE placentas and its overexpression led to an inhibition of syncytiotrophoblast differentiation, targeting key molecules in the process [69]. Fu et al. showed that miR-517-5p is highly expressed in placenta samples of PE pregnancies, which decreased cell proliferative and invasive abilities of human choriocarcinoma cell line by inhibiting extracellular signal regulated kinase/MMP-2 pathway [70]. Liu et al. showed that miR-518b can promote trophoblast cell proliferation via Rap1b–Ras–MAPK pathway, and the aberrant upregulation of miR-518b in preeclamptic placenta may contribute to the excessive trophoblast proliferation [71]. Mong et al. have shown that C19MC miRNAs have a crucial role in regulating epithelial-to-mesenchymal transition (EMT) genes in villous trophoblasts and maintaining their stem-like epithelial cell phenotype. The hypoxic condition during early placentation reduced C19MC expression and released the inhibition of EMT genes leading to the acquisition of migratory and invasive characteristics of EVTs. The authors conclude that maintaining optimal expression levels of C19MC is critical for EVT differentiation and invasion, while dysregulation of C19MC may result in impaired invasion associated with either the shallow placentation of PE or the exuberant invasion of placenta accrete [72].

MiR-517a/b or miR-517c were upregulated in PE placentas, and their upregulation in first trimester primary EVTs resulted in decreased trophoblast invasion and increased release of the anti-angiogenic protein soluble fms-like tyrosine kinase 1 (sFlt1) [73]. sFlt1 binds circulating angiogenic factors and consequently blocks their ability to induce angiogenesis, disturb normal signaling in maternal endothelium, in that way having a major impact in PE pathogenesis [74,75,76]. Upregulation of both miRNAs in first trimester primary EVTs also led to upregulation of tumor necrosis factor superfamily15 (TNFSF15), a cytokine involved in Flt1 splicing [73]. Additionally, it was demonstrated that syncytiotrophoblast-derived EVs directly transferred functional placental miRNA belonging to C19MC (mir-517a, mir-517c, mir-519a) to primary human endothelial cells, which may directly affect the maternal and fetal endothelial function [77]. Interestingly, Strub et al. showed that C19MC miRNA members could be involved in the aberrant angiogenesis in infantile hemangioma, which goes in line with the possible role that C19MC miRNA members may have in angiogenesis during pregnancy [78].

These results clearly indicate that C19MC miRNAs are important factors in controlling differentiation, migration and invasion of trophoblast cells as well as angiogenesis. Their dysregulation could lead to the impaired function of the trophoblast cells, contributing to impaired remodeling of the spiral arteries and consequently PE development.

### 2.3. Placenta Derived Extracellular MiRNAs and Immunomodulation

Placental miRNAs are released into the extracellular space and reach maternal circulation, where they are found in a very stable form of microparticles [53]. The main source of these circulating miRNAs are trophoblast cells [40,51,53]. These molecules can be released as microvesicles, exosomes, apoptotic bodies and as protein-bound miRNAs, (miRNAs complexed with Ago2, nucleophosmin1, or high-density lipoproteins (HDL)) [79,80,81]. Microvesicles originate from direct shedding of the plasma membrane; while on the other hand, release of exosomes is a regulated process, composed out of several steps which ultimately define the composition of the cargo within the vesicles. The release of miRNAs from cells in nonvesicular form is presumably an ATP-dependent process which also involves cell necrosis [40,81]. It is important to underline that extracellular miRNAs are not merely a simple measure of waste of cellular metabolism, apoptosis or necrosis; they act as a close or distant cell to cell communication toll. The extracellular miRNAs can be delivered to target cells and act as autocrine, paracrine, and/or endocrine regulators in that way potentially having a great impact on the function of the recipient cells [39,82]. Exosomes carrying specific miRNA cargo can be taken up by recipient cells via endocytic pathways or via direct fusing of the exosome particle with the cell membrane, with the release of their cargo into the cytoplasm [40,83] while on the other hand, miRNAs bound to proteins could interact with recipient cells via ligand–receptor interactions. Extracellular miRNAs can bind to Toll-like receptors and activate downstream signaling events, in that way regulating cell-to-cell communications [84]. MiRNAs that are carried in the circulation bound to HDL can be delivered to recipient cells via the HDL receptor scavenger receptor B type I [80].

Exosome vesicles, secreted from primary human trophoblasts (PHT) cells are packed with miRNAs, resembling the profile of trophoblastic cellular miRNA. The most abundant of them originate from C19MC [51]. It has been shown by several studies that C19MC derived miRNAs in these vesicles, can influence maternal immune cells and have a major impact in immunomodulation during pregnancy and defense against viral infection. Delorme-Axford et al. have provided evidence that PHT are highly resistant to infection by a number of viruses and more interestingly that they have an antiviral effect on other, non-placental cell types, by exosome-mediated delivery of specific miRNAs of the C19MC. They have shown that ectopically expressing the entire C19MC fragment in cells or transfecting a high level of certain C19MC members (miR517-3p, miR-512-3p, or 516b-5p) allocated viral resistance to the recipient non-placental cells by induction of autophagy [85]. Bayer et al. also demonstrated that PHT cells exert antiviral activity by at least two independent mechanisms, mediated by C19MC miRNA and by type III interferons [86], while Krawczynski et al. have found that miR-517a targets unc-13 homolog D (autophagy-related gene), suppressing in that way replication of vesicular stomatitis virus [87]. Furthermore, miR-517a-3p carried by placental exosomes regulates activation and proliferation of maternal T cells and NK cells, targeting the PRKG1 gene involved in activation of nitric oxide/cGMP signaling pathway [88]. Chaiwangyen et al. showed that miR-519d-3p released via extracellular vesicles from the trophoblast, can be taken up by other trophoblast cells and maternal immune cells and regulate their proliferation and migration, which may contribute to the immune tolerance in pregnancy [89]. All this evidence supports the specific role of miRNAs belonging to C19MC in regulation of the maternal immune system during pregnancy. The major part in pathogenesis of PE is related to immune response disorders and inflammation, triggered by humoral factors released from placenta and whole particles sequestered from the surface of the syncytium of the human placenta [10]. Therefore, it is possible that trophoblast derived exosomes carrying a potentially aberrant miRNA repertoire could severely influence maternal immune cells, leading to their dysfunction and in that way contribute to maternal systemic inflammatory response and the development and progression of PE [19,85,86,87,88]. The importance of the effect of miRNA cargo released from trophoblast cells in microparticles was underlined in the recent study by Hiu et al. [90]. The authors showed that trophoblastic small EV (sEV) are internalized into placental fibroblasts or uterine endothelial cells by macropinocytosis and clathrin-mediated endocytosis and demonstrated the trafficking of sEVs through the endosome-lysosome system and the delivery of sEV miRNA cargo to P-bodies. This represents the first data to suggest the delivery of sEV miRNAs to the RISC complex proteins, cellular site for miRNA-dependent silencing in the target cells [90] and strengthens the evidence of the impact of the microparticle miRNA cargo on cells participating in PE pathogenesis, both maternal and fetal origin.

### 2.4. Circulating MiRNAs as Biomarkers of PE

As already stated above, a specific placental miRNA pattern dynamically changes during pregnancy and those changes are reflected in maternal circulation [40,91,92]. Placenta derived EV are detected as early as six weeks of gestation, while after termination of pregnancy, plasma levels of C19MC associated miRNAs decrease significantly [93,94]. In contrary to cellular RNA species, the presence of miRNAs in vesicles or with accompanying proteins protects extracellular miRNAs from degradation and increases their stability in the extracellular milieu [39,95], making them a good biomarker candidate for the pathology of placenta. The additional fact in favor of this is that the release of microRNAs from the placenta mainly occurs from the villous trophoblast cells, indicating that circulating microRNAs could serve as unique markers for monitoring trophoblast and placental function [40,51,53]. Given the nature of miRNAs belonging to the C19MC cluster, their exclusive placenta-specific pattern of expression, high levels of expression in trophoblasts and presence in mother circulation, in vitro evidence for influencing crucial steps in PE pathogenesis, they have attracted great interest in PE biomarker research, both as prediction and diagnostic markers of the disease [45,53,85,94].

Many studies have shown abnormal levels of C19MC derived miRNAs in maternal circulation in pregnancies affected by PE, in different stages of the disease [64,68,70,96,97,98,99,100,101,102]. Several excellent reviews and a meta-analysis were written, giving us a useful overview of the potential of miRNA as prognostic and diagnostic tools in PE [29,30,103]. Recently, the miRNA analysis of plasma exosomes, either from placenta-specific isolated exosomes, or from whole plasma exosomes, has attracted much attention [17,19,20,104]. This approach could provide better sensitivity for the prediction of PE, however, cost effectiveness is a major issue. Even a decade later, great discrepancies still exist between the studies, with minimal overlap between the individual miRNAs identified as potential biomarkers Table 1 [17,29,30,96,97,98,99,100,101,102,103,104].

Nevertheless, in vitro studies along with the measurements of C19MC miRNAs in maternal blood provide evidence for the dysregulation of C19MC miRNAs during PE and their involvement in the regulation of crucial processes required for adequate placentation and immunomodulation during pregnancy. Their dysregulation could have a major effect in both stages of the disease.

### 2.5. Preanalytics and Technology

There are several major preanalytical and analytical points that should be considered when quantifying circulating miRNAs, which could be a cause of inconsistencies between the studies. When analyzing placenta derived non-specific miRNAs (which originate from other cells as well) in maternal circulation, different processing of the samples (time frame form venipuncture to the sample preparation and freezing) can have a major impact on the sample miRNA profile. The study of Mitchell et al. showed that freeze–thaw cycles, to a great extent, influence plasma miRNA profile, if not prepared as platelet poor plasma [105]. The reason for this is that non-specific placental miRNAs could be released from the residual blood cells, and in that way contaminate the sample [106]. These issues should also be considered in exosomal analysis of the plasma. Furthermore, serum and plasma have a different spectrum of miRNAs, due to the coagulation process itself, and the subsequent miRNA release, the process that we certainly cannot control in any sample [107]. Indeed, analyte which takes part in the coagulation process or changes during it is never quantified in serum samples, but in plasma instead. The logic here should be the same.

Regarding both placenta specific and nonspecific miRNAs, other causes of discrepancies could be due to cohort characteristics, gestational age (given that changes within placenta trough gestation are reflected in mothers’ circulation), PE form (EO or LO) [7,40,91,92]. Additionally, a great source of variation could be due to the use of different profiling high-throughput technologies (different microarray and next generation sequencing (NGS) platforms) [108,109,110,111,112,113,114,115].

As a first step in biomarker discovery, it is always wise to use some of the screening technologies. Earlier, some form of array technology was used: whether it was microarray [108] or qPCR arrays [109]. Microarrays were a good starting point. With time, more dense microarrays were designed, which allowed better target coverage. Its main challenge was normalization and buffering the potential bias between the samples and measurement variations which sometimes resulted in bad reproducibility of the results [110]. qPCR-arrays were later quite often used as a technique of screening, but still only a limited number of targets could be measured with this technique, since the qPCR-arrays are often thematic and cover dedicated signaling pathways [109]. With the development of new generation sequencing (NGS), or sometimes called deep sequencing, a new high-throughput technology emerged that allowed miRNA high-sensitive expression profiling without previous target selection. This technique enabled identification of novel miRNAs: revealing multiple miRNA variants with heterogeneous ends, lengths and expression levels [111,112]. In comparison to qPCR-arrays, NGS provides a cost-effective method in relation to high capacity. However, it requires a high input amount of the sample which could represent a technical challenge when circulating miRNAs are considered as potential biomarkers. There are available low-input miRNA library preparation kits, but the choice should be carefully evaluated for each sample type [113]. In addition, special attention must be paid to deal with biases associated with library preparation: adapter dimer formation, size selection of small RNA species, etc. [114,115]. In a study by Hui et al., of four maternal plasma exosome miRNAseq libraries, C19MC miRNAs were reliably detected in only one library, while numerous C19MC miRNAs were detected when qPCR-based TaqMan cards were used, questioning weather sequencing is sensitive enough for the detection of C19MC in plasma [19].

All these issues complicate the identification of a reliable biomarker for PE. In our research, we have used the digital droplet PCR (ddPCR) method for circulating miRNA quantification and shown that placenta specific miR-518b could serve as a potential biomarker for discriminating preeclampsia and healthy pregnancy [102]. DdPCR is a sensitive technique that enables absolute nucleic acid quantification, therefore represents a right tool for quantification of circulating miRNAs from low abundant samples. This technique is suitable for measuring a limited number of targets and is mostly used as a confirmation technology for the identified targets detected with screening techniques. It has many potential advantages over real-time PCR: obtaining higher precision and sensitivity, absolute quantification without the use of external references, with the advantage of removal of PCR efficiency bias [116,117]. It would be also the method of choice for the diagnostic use of miRNA biomarkers, since quantification by ddPCR in comparison to real-time PCR revealed greater precision (coefficients of variation decreased 37–86%) and improved day-to-day reproducibility of ddPCR [117]. More importantly, when applied to microRNA detection in clinical serum samples, ddPCR showed superior diagnostic performance compared to real-time PCR [117]. By comparing plasma exosome and whole plasma analysis of miRNAs in PE patients vs. controls, only one out of seven differentially expressed miRNAs in exosomes was detected in whole plasma samples [19]. This could be a consequence of the overall dilution of exosomes carrying placental miRNA in plasma. In this case, perhaps the application of ultra-sensitive ddPCR technology could be useful in overcoming these issues.

Small size and large number of closely related family members of miRNAs with highly similar sequences, makes it challenging to measure each miRNAs species separately accurately. Furthermore, several miRNAs at C19MC (e.g., the miR-520 and miR-519 families) have the similar “AAGUGC” seed sequence which is also found in members of the 371/miR-373 cluster and other miRNAs and nearly identical to that of the miR-17-92 cluster (also referred to as OncomiR1) [50]. Malnou et al. raised the question whether some miRNAs of C19MC sharing the same seed sequence may have related mRNA targets (or families thereof), perhaps underlying similar functions [49]. However, this question should also reflect the field of biomarker research, focusing perhaps on quantification of a panel of miRNAs with the same seed sequence or targets, especially taking into account that miRNAs from the C19MC cluster originate from the one large pre-miRNA transcript, and that the main regulation is its transcription [50,56,57].

## 3. Long Non-Coding RNAs

Long non-coding RNAs (lncRNAs) represent structurally and functionally a very diverse group of RNAs defined as transcripts over 200 nucleotides long characterized by poor conservation of primary sequences across species, low abundance in cells, high level of tissue specificity, and affinity to form secondary structural domains crucial for their functions [118,119]. Depending on the genomic loci of their origin, lncRNAs are classified as long intergenic, intronic, sense, antisense, promotor-associated, enhancer, and bidirectional [120]. Most lncRNAs are synthesized similarly to mRNAs through the activity of RNA polymerase II (RNAP II) [121]. In addition, RNAP III catalyzes the transcription of natural antisense transcripts from the antisense strand of exons, enhancer lncRNAs without a poly-A tale and intronic lncRNAs in combination with spRNAP IV [120,122]. LncRNAs also undergo intensive post-transcriptional modifications including 5’-capping, the addition of the poly-A tail, RNA editing and alternative splicing [120].

LncRNAs can regulate cellular processes through a variety of mechanisms. In the nucleus, lncRNAs can act through chromatin modifications, transcriptional regulations or interaction with enhancers. In the cytoplasm, lncRNAs act as miRNA sponges, influence mRNA stability and level of translation [123]. LncRNAs also play a role in regulating the expression of genes by acting as signals (participating in transmission of specific signaling pathways as a signal transduction molecule), scaffolds (downstream effectors can bind to the same lncRNA molecule to achieve information exchange and integration between different signaling pathways), guides (lncRNA can recruit chromatin-modifying enzymes to target genes) and decoys (lncRNA binds directly to proteins thus blocking the role of the molecule and signaling pathways) [124]. The biogenesis and functions of lncRNAs are described in Figure 2. Interestingly, some of the latest data suggested the existence of small open reading frames within lncRNAs with the potential to encode micropeptides, thus questioning the non-coding nature of lncRNAs [125]. Although the full biological roles of most lncRNA remain unknown, their roles in regulation of epigenetic and transcriptional mechanisms linked to PE are evident.

### 3.1. LncRNAs in PE

Several studies have used a transcriptomic approach to identify lncRNAs involved with PE. Through microarray analysis and RNA profiling, He et al. identified 738 out of 28,443 differentially expressed lncRNAs in the placentas of PE in comparison to normal pregnancies [126]. Additionally, they have examined the expression of three lncRNAs (LOC391533, LOC284100, CEACAMP8) in 40 preeclampsia placenta tissues and 40-matched control placenta tissues using qPCR and proposed their involvement in PE through the regulation of angiogenesis and vasculogenesis (via vascular endothelial growth factor receptor 1) and lipid metabolism (via lipoprotein lipase) [126]. In another study, Liu S. et al. using combined microarray and RNA-seq approach identified several differentially expressed lncRNAs linked to PE and JAK-STAT signaling pathway activation implicated in the progression of PE [127]. By performing RNA-seq on decidual samples, Tong et al. discovered that 32 lncRNAs were differentially expressed between normal and early-onset PE, 53 lncRNAs were differentially expressed between normal and late-onset PE, and 32 differentially expressed lncRNAs between early-onset PE and late-onset PE, suggesting that different pathophysiological mechanisms are driving early-onset and late-onset PE [128]. In the study focusing on the lncRNAs role in the development of early-onset PE, Wang et al. identified 15,646 upregulated and 13,178 downregulated lncRNAs by microarray in the placenta of EOPE patients compared to the preterm controls. Through additional GO analysis, the authors showed that pathways overrepresented in the EOPE patients were related to cell migration and cell motility [129].

So far, particular roles of several lncRNAs in PE development and progression have been described. Upregulation and downregulation of specific lncRNAs can impact critical mechanisms in the development of PE and lead to changes in trophoblast proliferation, invasion, migration and apoptosis Table 2 [130,131,132,133,134,135,136,137,138,139,140,141,142,143,144].

### 3.2. Circulating LncRNAs in PE

So far, several lncRNA in the circulation have been explored as potential biomarkers for PE. Sun et al. showed that levels of lncRNA BC030099 in whole blood had a good ability to discriminate preeclamptic from healthy pregnant women (AUC = 0.713). Furthermore, lncRNAs NR_026824.1, AK055151.1, NR_027457, and NR_024178 were highly upregulated in PE placentas but did not demonstrate good biomarker potential for PE (AUC < 0.6) [145]. Luo et al. suggested that levels of three lncRNAs in serum AF085938, G36948, and AK002210 can also serve as potential diagnostic biomarkers in preeclampsia (AUC = 0.7673, 0.7956, and 0.7575, respectively) [146]. Dai et al. have identified seven lncRNA in early pregnancy as potential biomarkers for the prediction of pregnancy-induced hypertension and preeclampsia. However, in ROC analysis, these seven lncRNAs (NR_002187, ENST00000398554, ENST00000586560, TCONS_00008014, ENST00000546789, ENST00000610270, and ENST00000527727,) showed only modest discriminative power (AUC between 0.6 and 0.7) [147].

Currently, the main problems in the quantification of circulating lncRNAs are related to their low abundance in the circulation and unstandardized strategies in normalization approaches. Many authors utilize mRNA levels of housekeeping genes as a reference in normalization. However, a significant difference in the stability of circulating mRNAs compared with lncRNAs questions the validity of this approach [148]. In the study by Dong et al., β-actin was shown to be the most stable in comparison to GAPDH, HPRT, 18S RNA, CYC, and GUSB in the serum of healthy and cancer patients [149]. The other approach to normalization includes the use of artificial spike-ins that enable good quality control of analytical variations (extraction, efficacy of reverse transcriptions and qPCR). On the other hand, the spike-ins are synthetic oligomers externally added to the sample, and as such have limited capacity to control preanalytical factors like sampling, hemolysis, etc. [150]. It seems important to reach a consensus regarding current dilemmas to exploit the full biomarker potential of circulating lncRNAs.

## 4. Conclusions

PE is characterized by extensive dysfunction of the placenta, caused by dysregulation of trophoblast differentiation, invasion and ultimately remodeling of the spiral arteries [4,5,6]. The disturbances in placentation have major consequences later on in pregnancy, causing extensive systemic inflammatory response which has a major impact on maternal and fetal health [7,8,9]. The results from many in vitro studies clearly indicate that miRNAs belonging to C19MC are important factors in controlling crucial processes required for adequate placentation, and that dysregulation of C19MC could lead to the impaired function of the trophoblast cells, impaired placentation and consequently PE development [66,67,68,69,70,71,72,73,74,75,76,77,78]. In addition, upregulation and downregulation of specific lncRNAs can also impact trophoblast proliferation, invasion, migration as well as apoptosis [130,131,132,133,134,135,136,137,138,139,140,141,142,143,144].

Considering miRNAs belonging to C19MC are involved in the regulation of maternal immune system during pregnancy, it is possible that in PE, trophoblast derived exosomes carrying a potentially aberrant miRNA repertoire could severely influence maternal immune cells, contributing to maternal systemic inflammatory response and progression of PE [19,85,86,87,88]. All these data are supported by the evidence that miRNAs belonging to C19MC, secreted from trophoblast cells and carried by microparticles, are taken up recipient cells [40,85,88,89,90] and actually delivered to the RISC complex proteins of both maternal and fetal recipient cells [90].

Extensive involvement of miRNAs belonging to C19MC in PE pathogenesis [66,67,68,69,70,71,72,73,74,75,76,77,78,85,86,87,88], their detection in maternal circulation early in pregnancy and exclusive placenta-specific pattern of expression [40,49,93,94] marks them as promising prognostic and diagnostic tools for PE. However, major inconsistencies exist between the studies examining their diagnostic utility in PE analyzing miRNAs either from maternal serum/plasma or from plasma exosomes [68,70,96,97,98,99,100,101,102]. Circulating lncRNAs have also been considered as biomarkers of PE, however, their low abundance in the circulation and unstandardized strategies in normalization approaches also represent challenges for their clinical utility in PE [148,149]. Therefore, critical preanalytical (sample handling, sample type) and analytical issues (use of various quantification approaches with different sensitivity and specificity, normalization), regarding both miRNAs and lncRNA quantification, should be resolved prior to potential clinical application of these markers in PE assessment.

## Figures and Tables

**Figure 1 ijms-22-10652-f001:**
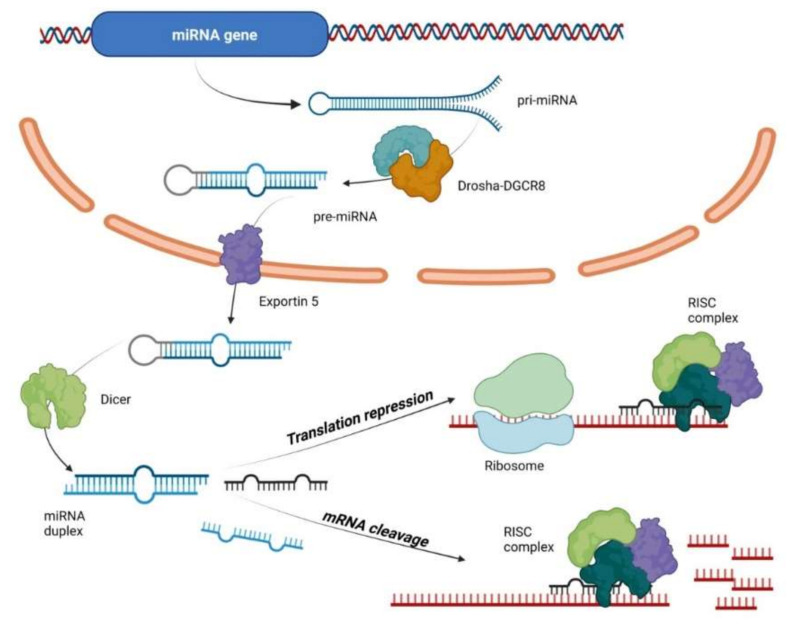
MicroRNA biogenesis and mechanism of action. Primary miRNAs (pri-miRNAs) are transcribed from their genes and processed into double-stranded precursors (pre-miRNAs), by the microprocessor complex: RNA binding protein DiGeorge Syndrome Critical Region 8 (DGCR8) and Drosha. Pre-miRNAs are exported to the cytoplasm by an exportin 5 (XPO5)/RanGTP complex and processed by the RNase III endonuclease Dicer yielding a mature miRNA duplex. This step is followed by unwinding of mature miRNA duplex and loading into the Argonaute2 (Ago2)-containing, RNA-induced silencing complex (RISC) where the miRNAs repress mRNAs in a sequence-specific manner.

**Figure 2 ijms-22-10652-f002:**
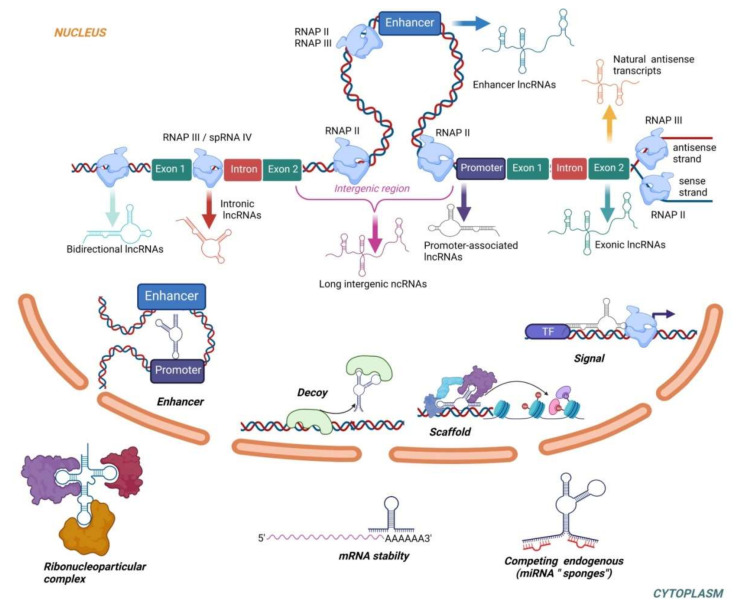
The biogenesis and functions of lncRNAs. Based on the transcriptional origin, lncRNAs are classified as exonic, intronic, antisense, enhancer, intergenic, promoter-associated and bidirectional. The vast majority of lncRNAs is transcribed by RNAP II. RNAP III catalyzes transcription of natural antisense transcripts from the antisense strand of exons, enhancer lncRNAs without poly-A tale and intronic lncRNAs in combination with spRNAP IV. In the nucleus, lncRNAs regulate gene expression by acting as transcriptional signals, scaffolds, decoys or enhancers. In the cytoplasm, lncRNAs can increase/decrease mRNA stability, act as miRNAs “sponges” or interact with proteins to form ribonucleoprotein complexes.

**Table 1 ijms-22-10652-t001:** C19MC miRNA differential expression in maternal circulation of PE patients.

MiRNA	Sample	Expression	Technology	Time of Sampling	Study Type	Year
miR-516-5p, miR-517 *, miR-520a *, miR-525, miR-526a	Maternal plasma	Upregulated	qPCR	Third trimester	PE diagnosis	2013 [96]
miR-517c, miR-518-3p, miR-518e, miR-519d	Maternal plasma	Upregulated	SOLiD™ sequencing	At delivery	PE diagnosis	2015 [97]
miR-518b, miR-1323, miR-516b, miR-516a-5p, miR-525-5p, miR-515-5p, miR-520 h, miR-520a-5p, miR-519d, miR-526b	Maternal plasma	Upregulated	qPCR	27–34 WG ^+^	Severe PE diagnosis Early onset; late onset vs. controls	2015 [98]
miR-517-5p, miR-518b, miR-520h	Maternal plasma	Upregulated	qPCR	First trimester	PE prediction	2017 [99]
MiR-520g	Maternal serum	Upregulated	qPCR	First trimester	PE prediction	2017 [68]
miR-517c-3p	Maternal plasma	Upregulated	qPCR	Third trimester	PE diagnosis	2018 [70]
miR-520c-3p, miR-518f, miR-512-3p, miR-520d-3p	Maternal serum	Upregulated	TaqMan low density array plates	12, 16, 20 WG ^+^	PE prediction	2018 [100]
miR-517-5p, miR-520a-5p, miR-525-5p	Plasma exosomes	Downregulated	qPCR	First trimester	PE and gestational hypertension prediction	2019 [101]
miR-518b	Maternal plasma	Upregulated	ddPCR	Third trimester	PE diagnosis	2020 [102]

* According to the previous nomenclature [96]. ^+^ WG—weeks of gestation.

**Table 2 ijms-22-10652-t002:** Dysregulation of specific lncRNAs in PE and proposed mechanisms of action.

	Differential Expression in Placenta	Target	Function
uc.187	Upregulation	PCNA/Ki67 and caspase-3/Bcl-2	The upregulation of lncRNA uc.187 modulates PCNA/Ki67 and caspase-3/Bcl-2 pathways leading to inhibition of cell proliferation, invasion, and increased cell apoptosis [130].
SPRY4-IT1	Upregulation	Wnt/β-catenin pathway	The upregulation of SPRY4-IT1 decreases migration and proliferation and increases apoptosis of extravillous trophoblast cells by acting through the Wnt/β-catenin pathway [131].
RPAIN	Upregulation	-MMP2 and MMP9 C1q complement	RPAIN overexpression is followed by downregulation of MMP2 and MMP9 which leads to the reduction in trophoblast invasion [132]. RPAIN increases apoptosis of trophoblast through inhibition of C1q complement expression [132].
CCAT1	Upregulation	Cyclin D1-P16-CDK4 pathway	The CCAT1 upregulation decreases trophoblast proliferation through inhibition of cyclin D1-P16-CDK4 pathway [133].
HOTAIR	Upregulation	miR-106	Upregulation of HOTAIR represses proliferation, migration and invasion of trophoblast cells by decreasing miR-106 expression through EZH2-dependent methylation of promoter [134].
DC	Upregulation	p-STAT3 pathway	The increased expression of lncRNA DC induces over-maturation of decidual dendritic cells through the p-STAT3 pathway, which leads to the expansion of Th-1 cells and contributes to a continuous inflammatory state in preeclampsia patients [135].
H19	Downregulation-Upregulation	miR-657 and NOMO1 signaling PI3K/AKT/mTOR pathways	Downregulation of H19 causes a decrease in miR-657 production and an increase in NOMO1 signaling leading to the excessive proliferation of trophoblast cells observed in early-onset PE [136]. H19 upregulation decreased the viability and promoted the invasion and autophagy in the trophoblast cells via activation of the PI3K/AKT/mTOR pathways [137].
ATB	Downregulation	?	Downregulation of ATB leads to decreased migration, proliferation, and tube formation of trophoblastic cells [138].
PVT1	Downregulation	PI3K/AKT pathway ANGPTL4	Downregulation of PVT1 causes a decrease in proliferation and migration of trophoblast cells through the PI3K/AKT pathway [139]. Decreased binding of PVT1 to EZH2 in trophoblast cells leads to inhibition of cell proliferation and stimulation of cell cycle accumulation and apoptosis through upregulation of ANGPTL4 [140].
TUG1	Downregulation	RND3 miR-29b miR-29a-3p	By interacting with EZH2, TUG1 reduces transcription of RND3 and promotes the proliferation, invasion, migration and apoptosis of trophoblasts, as well as spiral artery remodeling [141]. Sponging of miR-29b leads to downregulation of MCL1, VEGFA, and MMP2 and modulates proliferation, apoptosis, invasion, and angiogenesis in trophoblast cells [142]. Sponging of miR-29a-3p, activates VEGFA and Ang2/Tie2 signaling, again facilitating trophoblast cell proliferation, migration, invasion, and angiogenesis [143].
MVIH	Downregulation	Jun-B protein	The downregulation of MVIH modulates Jun-B protein expression which leads to the inhibition of trophoblast cell growth, migration, and invasion [144].

Ang2/Tie2—angiopoietin receptors 2; NOMO1—nodal modulator 1; PI3K—phosphoinositide 3-kinases; AKT—protein kinase B (PKB); mTOR—mechanistic target of rapamycin; ANGPTL4—angiopoietin-like 4; RND3—Rho family of GTPases 3; EZH2—enhancer of zeste homolog 2; MCL1—induced myeloid leukemia cell differentiation protein; VEGFA—vascular endothelial growth factor A; PCNA—Proliferating Cell Nuclear Antigen; Ki-67—Marker Of Proliferation Ki-67; Bcl-2—B-cell lymphoma 2; MMP2—matrix metalloproteinase 2; MMP9—matrix metalloproteinase 9; D1—cyclin D1; P16—cyclin-dependent kinase inhibitor 2A; CDK4—cyclin-dependent kinase 4; STAT3—signal transducer and activator of transcription 3.

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
