# Peer review of "Non-Coding RNAs in Preeclampsia—Molecular Mechanisms and Diagnostic Potential"

_ijms, 2021, doi:10.3390/ijms221910652_

Round 1

Reviewer 1 Report

The review manuscript titled “Non-coding RNAs in preeclampsia - molecular mechanisms and diagnostic potential” by Dr. Munjas and colleagues provides a detailed overview on the role/currently known molecular mechanisms of non-coding RNAs, including miRNAs and lncRNAs (with particular focus on the miRNA cluster C19MC) in the pathogenesis of preeclampsia. The current state of the art about the use of these RNA molecules as predictive biomarkers is also provided.

The review is interesting, well written in general and relatively easy to follow. Overall, this topic is quite important and interesting. This review can help to better understand the role of ncRNAs dysregulation in preeclampsia. While I recommend the article for publication, I have some observations to be reviewed by authors before such publication:

Main points
1.    May I suggest removing the word “review” from the title? The title as it is is giving the redundant information that the ms is a review
2.    Two brief paragraphs and figures describing the ncRNA/miRNA biogenesis would be helpful for the reader
3.    Spaces between paragraphs should be removed. For instance, between lines 54-55, 69-70, 72-73 etc..in addition, several subheads should be separated with a space from the previous pharagraph, eg., lines 37-38, line 152-153, lines 214-215 and others
4.    Please include the aim of the review at the end of the introduction section
5.    As the 2.3 section is describing the function of miRNAs belonging to C19MC cluster, and the 2.1 is describing the general molecular characteristics/function of that miRNA cluster, I suggest switching the 2.3-2.2 sections. It might be easier to follow the text

Minor observations
Lines 22 miRNAs and lncRNAs. Should be microRNA (miRNA) and long non-coding RNA (lncRNAs) when mentioned for the first time. 
Line 27 Is C19MC a lncRNA or a miRNA? This information should be detailed in the abstract
Line 113 C19MC cluster?
Line 116 differentiated trophoblast cells? 
Line 118 certain miRNAs from C19MC cluster?
Line 136 please define the meaning of adjacent. Up-stream or downstream?
Lines144-146 please include reference. For instance PMID: 28487016
Line 158 please remove the double parentheses 
Line 245 I suggest including abbreviations within parenthesis as the other sections of the ms “soluble fms-like tyrosine kinase 1s (Flt1)” and maintain the same style. 
Line 264 (48, 83, 84). The reference style should be uniformed
Lines 279-280 Please include references. For instance PMID: 28487016
Line 322-327 Supporting references should be included
Line 386 it would be better as “Long non-coding RNAs”
Lines 387-398 The lncRNA paragraph is lacking in supporting references. For instance, a detailed description of the molecular characteristics and functional role of lnRNAs is detailed here (PMID: 33898434). In addition, the “sponge” function is only one of the currently known lncRNAs functions in regulating gene expression. LncRNAs also play a role in regulating the expression of genes by acting as as signal, scaffold, guide, and decoy. These notions are described in detail here (PMID: 2496520, PMID: 31048188). I suggest including these important information/supporting refs
Line 404 Have LOC391533, 404 LOC284100 and CEACAMP8 been functionally validated in vitro or also in vivo?
Line 423-430 Please revise the format of the abbreviations 
Line 465 quoted papers should be included as references
Line 472 Better conclusions? 

Author Response

Reviewer 1

The review manuscript titled “Non-coding RNAs in preeclampsia - molecular mechanisms and diagnostic potential” by Dr. Munjas and colleagues provides a detailed overview on the role/currently known molecular mechanisms of non-coding RNAs, including miRNAs and lncRNAs (with particular focus on the miRNA cluster C19MC) in the pathogenesis of preeclampsia. The current state of the art about the use of these RNA molecules as predictive biomarkers is also provided.

The review is interesting, well written in general and relatively easy to follow. Overall, this topic is quite important and interesting. This review can help to better understand the role of ncRNAs dysregulation in preeclampsia. While I recommend the article for publication, I have some observations to be reviewed by authors before such publication:

We would like to thank the reviewer for hers/his thoughtful reading and helpful comments on the manuscript. We went through suggestions and here are our answers:

Main points

  1.    May I suggest removing the word “review” from the title? The title as it is is giving the redundant information that the ms is a review

The authors removed “review” from the title.

2.    Two brief paragraphs and figures describing the ncRNA/miRNA biogenesis would be helpful for the reader

We have added two brief paragraphs and 2 figures to the article:

MicroRNAs

‘’Half of all currently identified miRNAs are transcribed mostly from introns and few exons of protein coding genes, while the remaining are intergenic, transcribed independently of a host gene [31, 32]. Their biogenesis starts with the processing of RNA polymerase II/III transcripts post- or co-transcriptionally [33]. In the canonical, dominant pathway of miRNA biogenesis, primary miRNAs (pri-miRNAs) are transcribed from their genes and processed into double-stranded precursors (pre-miRNAs), by the microprocessor complex: RNA binding protein DiGeorge Syndrome Critical Region 8 (DGCR8) and a ribonuclease III enzyme, Drosha [34]. DGCR8 recognizes an N6-methyladenylated GGAC and other motifs within the pri-miRNA [35], while Drosha cleaves the pri-miRNA duplex at the base of the characteristic hairpin structure of pri-miRNA. Pre-miRNAs are exported to the cyto-plasm by an exportin 5 (XPO5)/RanGTP complex and further processed by the RNase III endonuclease Dicer [34, 36], which results in the removal of the terminal loop, yielding a mature miRNA duplex [33, 37]. These double-stranded sequences are loaded into the Argonaute2 (Ago2)-containing, RNA-induced silencing complex (RISC) [38], where the miRNAs repress mRNAs in a sequence-specific manner [39]. The miRNA biogenesis pathway is shown in Figure 1. In addition, multiple non-canonical miRNA biogenesis pathways exist, using different combinations of the proteins involved in the canonical pathway, mainly Drosha, Dicer, exportin 5, and AGO2 [40].’’

Figure 1. MicroRNA biogenesis and mechanism of action. Primary miRNAs (pri-miRNAs) are transcribed from their genes and processed into double-stranded precursors (pre-miRNAs), by the microprocessor complex: RNA binding protein DiGeorge Syndrome Critical Region 8 (DGCR8) and Drosha. Pre-miRNAs are exported to the cytoplasm by an exportin 5 (XPO5)/RanGTP complex and processed by the RNase III endonuclease Dicer yielding a mature miRNA duplex. This step is followed by unwinding of mature miRNA duplex and loading into the Argonaute2 (Ago2)-containing, RNA-induced silencing complex (RISC) where the miRNAs repress mRNAs in a sequence-specific manner.

Long non-coding RNAs

‘’ Depending on the genomic loci of their origin, lncRNAs are classified as long intergenic, intronic, sense, antisense, promotor-associated, enhancer, and bidirectional [121]. Most lncRNAs are synthesized similarly to mRNAs through the activity of RNA polymerase II (RNAP II) [122]. In addition, RNAP III catalyzes transcription of natural antisense transcripts from the antisense strand of exons, enhancer lncRNAs without poly-A tale and intronic lncRNAs in combination with spRNAP IV. [121, 123]. LncRNAs also undergo intensive post-transcriptional modifications including 5'-capping, the addition of the poly-A tail, RNA editing, and alternative splicing [121].’’

Figure 2. The biogenesis and functions of lncRNAs. Based on the transcriptional origin lncRNAs are classified as: exonic, intronic, antisense, enhancer, intergenic, promoter-associated and bidirectional. The vast majority of lncRNAs is transcribed by RNAP II. RNAP III catalyzes transcription of natural antisense transcripts from the antisense strand of exons, enhancer lncRNAs without poly-A tale and intronic lncRNAs in combination with spRNAP IV. In the nucleus, lncRNAs regulate gene expression by acting as transcriptional signals, scaffolds, decoys or enhancers. In the cytoplasm, lncRNAs can increase/decrease mRNA stability, act as miRNAs “sponges” or interact with proteins to form ribonucleoprotein complexes. 

  1.    Spaces between paragraphs should be removed. For instance, between lines 54-55, 69-70, 72-73 etc..in addition, several subheads should be separated with a space from the previous pharagraph, eg., lines 37-38, line 152-153, lines 214-215 and others

We have made the requested changes.

  1.    Please include the aim of the review at the end of the introduction section

We have added to the introduction section: The aim of this review is to summarize the latest knowledge of the molecular mechanism by which miRNAs (focusing on the chromosome 19 miRNA cluster (C19MC) and lnRNAs contribute to pathophysiology of PE and to review the latest evidence of their potential utility as biomarkers of PE.

  1.    As the 2.3 section is describing the function of miRNAs belonging to C19MC cluster, and the 2.1 is describing the general molecular characteristics/function of that miRNA cluster, I suggest switching the 2.3-2.2 sections. It might be easier to follow the text

The sections have been switched.

Minor observations

Lines 22 miRNAs and lncRNAs. Should be microRNA (miRNA) and long non-coding RNA (lncRNAs) when mentioned for the first time. 

It has been added.
Line 27 Is C19MC a lncRNA or a miRNA? This information should be detailed in the abstract

C19MC is a miRNA cluster, it has been clarified now in the abstract.

„….focusing on the chromosome 19 miRNA cluster (C19MC)..“
Line 113 C19MC cluster?

The abbreviation of chromosome 19 miRNA cluster is C19MC
Line 116 differentiated trophoblast cells? 

The paper by Lee et all (ref. 44) states of primary first-trimester trophoblast, we have added the clarification.
Line 118 certain miRNAs from C19MC cluster?

We have added: “..The expression of certain miRNAs from C19MC..“
Line 136 please define the meaning of adjacent. Up-stream or downstream?

The authors have added “The miR-371-3 cluster is located on chromosome 19 within a 1050 bp region approximately 20 kb downstream of the adjacent to the C19MC”.

Lines144-146 please include reference. For instance PMID: 28487016

We have added the reference PMID: 28487016.
Line 158 please remove the double parentheses 

Removed.
Line 245 I suggest including abbreviations within parenthesis as the other sections of the ms “soluble fms-like tyrosine kinase 1s (Flt1)” and maintain the same style. 

We have made the requested changes..
Line 264 (48, 83, 84). The reference style should be uniformed

We have made the requested changes..
Lines 279-280 Please include references. For instance PMID: 28487016

We have added the supporting references.
Line 322-327 Supporting references should be included

We have added the supporting references.
Line 386 it would be better as “Long non-coding RNAs”

We have made the change.
Lines 387-398 The lncRNA paragraph is lacking in supporting references. For instance, a detailed description of the molecular characteristics and functional role of lnRNAs is detailed here (PMID: 33898434). In addition, the “sponge” function is only one of the currently known lncRNAs functions in regulating gene expression. LncRNAs also play a role in regulating the expression of genes by acting as as signal, scaffold, guide, and decoy. These notions are described in detail here (PMID: 2496520, PMID: 31048188). I suggest including these important information/supporting refs

We have added the supporting references and described additional functions of lncRNAs.
Line 404 Have LOC391533, 404 LOC284100 and CEACAMP8 been functionally validated in vitro or also in vivo?

We have modified the sentence to make it clear that lncRNAs have been validated in the placental tissues ’’ Additionally, they have examined the expression of three lncRNAs (LOC391533, LOC284100, CEACAMP8) in 40 preeclampsia placenta tissues and 40-matched control placenta tissues using qPCR..’’
Line 423-430 Please revise the format of the abbreviations 

We have made changes to the format of the abbreviations.
Line 465 quoted papers should be included as references

We have added the following references:

Guglas K, Bogaczyńska M, Kolenda T, Ryś M, Teresiak A, Bliźniak R, Łasińska I, Mackiewicz J, Lamperska K. lncRNA in HNSCC: challenges and potential. Contemporary Oncology. 2017;21(4):259.

Dong L, Qi P, Xu MD, et al. Circulating CUDR, LSINCT-5 and PTENP1 long noncoding RNAs in sera distinguish patients with gastric cancer from healthy controls. Int J Cancer 2015; 137: 1128-35.
Line 472 Better conclusions? 

We have changed the subtitle, and extended the conclusion section.

PE is characterised by extensive dysfunction of the placenta, caused by dysregulation of trophoblast differentiation, invasion and ultimately remodelling of the spiral arteries [4-6]. The disturbances in placentation have major consequences later on in pregnancy, causing extensive systemic inflammatory response which has a major impact on maternal and fetal health [7-9]. The results from many in vitro studies clearly indicate that miRNAs belonging to C19MC are important factors in controlling crucial processes required for adequate placentation, and that dysregulation of C19MC could lead to the impaired function of the trophoblast cells, impaired placentation and consequently PE development [66-78]. In addition, upregulation and downregulation of specific lncRNAs can also impact trophoblast proliferation, invasion, migration as well as apoptosis [131-145].

Considering miRNAs belonging to C19MC are involved in the regulation of maternal immune system during pregnancy, it is possible that in PE, trophoblast derived exosomes carrying a potentially aberrant miRNA repertoire could severely influence maternal immune cells, contributing to maternal systemic inflammatory response and progression of PE [19, 85-88]. All these data are supported by the evidence that miRNAs belonging to C19MC, secreted from trophoblast cells and carried by microparticles, are taken up recipient cells [39, 85, 88-90] and actually delivered to the RISC complex proteins of both maternal and fetal recipient cells [90].

Extensive involvement of miRNAs belonging to C19MC in PE pathogenesis [66-78, 85-88], their detection in maternal circulation early in pregnancy and exclusive placenta-specific pattern of expression [39, 49, 93, 94] marks them as promising prognostic and diagnostic tools for PE. However, major inconsistencies exist between the studies examining their diagnostic utility in PE analysing miRNAs either from maternal serum/plasma or from plasma exosomes [68, 70, 96-102]. Circulating lncRNAs have also been considered as biomarkers of PE, however, their low abundance in the circulation and unstandardized strategies in normalization approaches also represent challenges for their clinical utility in PE[149, 150]. Therefore, critical preanalytical (sample handling, sample type) and analytical issues (use of various quantification approaches with different sensitivity and specificity, normalization), regarding both miRNAs and lncRNA quantification, should be resolved prior to potential clinical application of these markers in PE assessment.

Reviewer 2 Report

The review article discusses the potential role of C19MC miRNAs and lncRNAs in the pathogenesis of preeclampsia and their diagnostic utility, also the analytical challenges of non-coding RNA biomarker discovery.

The article itself is well designed, however, there are several smaller /bigger grammatical and phrasing errors throughout the article, which must be corrected.

I list some of these just for giving examples: 

line 81: "full biological role"

line 107: "solei"

line 112: "composed out of"

line 119: "germe cells"

line 129: "comprised out of"

line 168: "caring"

line 178: "originating form"

line 313: "taw"

line 317: "expression panels"

...and so on.

Other points that must be checked and corrected:

  • line 64: the size of the EVs is not "increased", only the number (release rate) and the cargo composition of the vesicles 
  • line 271: It is doubtful that the main form of miRNA release is the exosomal pathway as the majority of circulating miRNAs are not packaged within exosomes, but it is presumably the most functional pathway
  • The 3.1 and 3.2 sections have the same title.
  • line 467-469: Spike-ins are not appropriate for the control of the preanalytical variations (sampling, haemolysis, etc.), as these are added as external, synthetic factors to the samples, although, should be used for the QC of extraction, RT and PCR processes. For normalization, at least one internal control should be used, which is stable and has similar amplification characteristics to the target.
  • The conclusion is too short in view of the volume and content of the whole article.

Regards,

Author Response

The review article discusses the potential role of C19MC miRNAs and lncRNAs in the pathogenesis of preeclampsia and their diagnostic utility, also the analytical challenges of non-coding RNA biomarker discovery.

The article itself is well designed, however, there are several smaller /bigger grammatical and phrasing errors throughout the article, which must be corrected.

We would like to thank the reviewer for hers/his thoughtful reading and helpful comments on the manuscript. We went through suggestions and here are our answers:

I list some of these just for giving examples: 

line 81: "full biological role"

line 107: "solei"

line 112: "composed out of"

line 119: "germe cells"

line 129: "comprised out of"

line 168: "caring"

line 178: "originating form"

line 313: "taw"

line 317: "expression panels"

...and so on.

A native English speaker has reviewed and corrected the manuscript.

Other points that must be checked and corrected:

  • line 64: the size of the EVs is not "increased", only the number (release rate) and the cargo composition of the vesicles 

Thank you for the comment; we have removed “increased”

  • line 271: It is doubtful that the main form of miRNA release is the exosomal pathway as the majority of circulating miRNAs are not packaged within exosomes, but it is presumably the most functional pathway

Thank you for the comment, we have made changes to the sentence: “The additional fact in favour of this is that the release of microRNAs from the placenta mainly occurs from the villous trophoblast cells, indicating that circulating microRNAs could serve as unique markers for monitoring trophoblast and placental function.”

  • The 3.1 and 3.2 sections have the same title.

We have added to the title of section 3.2 Circulating lncRNAs in PE

  • line 467-469: Spike-ins are not appropriate for the control of the preanalytical variations (sampling, haemolysis, etc.), as these are added as external, synthetic factors to the samples, although, should be used for the QC of extraction, RT and PCR processes. For normalization, at least one internal control should be used, which is stable and has similar amplification characteristics to the target.

We have made the following changes according to the reviewers comments:

However, a significant difference in the stability of circulating mRNAs compared with lncRNAs questions the validity of this approach [149]. In the study by Dong et al, β-actin was shown to be the most stable in comparison to GAPDH, HPRT, 18S RNA, CYC, and GUSB in the serum of healthy and cancer patients [150].The other approach to normalization includes the use of artificial spike-ins that enable good quality control of analytical variations (extraction, efficacy of reverse transcriptions and qPCR). On the other hand, the spike-ins are synthetic oligomers externally added to the sample, and as such have limited capacity to control preanalytical factors like sampling, haemolysis etc [151].

  • The conclusion is too short in view of the volume and content of the whole article.
  • We have changed the subtitle, and extended the conclusion section

PE is characterised by extensive dysfunction of the placenta, caused by dysregulation of trophoblast differentiation, invasion and ultimately remodelling of the spiral arteries [4-6]. The disturbances in placentation have major consequences later on in pregnancy, causing extensive systemic inflammatory response which has a major impact on maternal and fetal health [7-9]. The results from many in vitro studies clearly indicate that miRNAs belonging to C19MC are important factors in controlling crucial processes required for adequate placentation, and that dysregulation of C19MC could lead to the impaired function of the trophoblast cells, impaired placentation and consequently PE development [66-78]. In addition, upregulation and downregulation of specific lncRNAs can also impact trophoblast proliferation, invasion, migration as well as apoptosis [131-145].

Considering miRNAs belonging to C19MC are involved in the regulation of maternal immune system during pregnancy, it is possible that in PE, trophoblast derived exosomes carrying a potentially aberrant miRNA repertoire could severely influence maternal immune cells, contributing to maternal systemic inflammatory response and progression of PE [19, 85-88]. All these data are supported by the evidence that miRNAs belonging to C19MC, secreted from trophoblast cells and carried by microparticles, are taken up recipient cells [39, 85, 88-90] and actually delivered to the RISC complex proteins of both maternal and fetal recipient cells [90].

Extensive involvement of miRNAs belonging to C19MC in PE pathogenesis [66-78, 85-88], their detection in maternal circulation early in pregnancy and exclusive placenta-specific pattern of expression [39, 49, 93, 94] marks them as promising prognostic and diagnostic tools for PE. However, major inconsistencies exist between the studies examining their diagnostic utility in PE analysing miRNAs either from maternal serum/plasma or from plasma exosomes [68, 70, 96-102]. Circulating lncRNAs have also been considered as biomarkers of PE, however, their low abundance in the circulation and unstandardized strategies in normalization approaches also represent challenges for their clinical utility in PE[149, 150]. Therefore, critical preanalytical (sample handling, sample type) and analytical issues (use of various quantification approaches with different sensitivity and specificity, normalization), regarding both miRNAs and lncRNA quantification, should be resolved prior to potential clinical application of these markers in PE assessment.
